# The Use of Cerebellar Hypoperfusion Assessment in the Differential Diagnosis of Multiple System Atrophy with Parkinsonism and Progressive Supranuclear Palsy-Parkinsonism Predominant

**DOI:** 10.3390/diagnostics12123022

**Published:** 2022-12-02

**Authors:** Natalia Madetko-Alster, Piotr Alster, Bartosz Migda, Michał Nieciecki, Dariusz Koziorowski, Leszek Królicki

**Affiliations:** 1Department of Neurology, Medical University of Warsaw, Kondratowicza 8, 03-242 Warsaw, Poland; 2Diagnostic Ultrasound Lab, Department of Pediatric Radiology, Medical University of Warsaw, ul. Kondratowicza 8, 03-242 Warsaw, Poland; 3Department of Radiology, National Institute of Geriatrics, Rheumatology and Rehabilitation, st. Spartańska 1, 02-637 Warsaw, Poland; 4Department of Nuclear Medicine, Medical University of Warsaw, ul. Banacha 1a, 02-097 Warsaw, Poland

**Keywords:** SPECT—single photon emission computed tomography, MSA-P, PSP-P, differential diagnosis, neuroimaging, atypical parkinsonian syndromes

## Abstract

The differential diagnosis of MSA-P and PSP-P remains a difficult issue in clinical practice due to their overlapping clinical manifestation and the lack of tools enabling a definite diagnosis ante-mortem. This paper describes the usefulness of SPECT HMPAO in MSA-P and PSP-P differentiation through the analysis of cerebellar perfusion of small ROIs. Thirty-one patients were included in the study—20 with MSA-P and 11 with PSP-P; the analysis performed indicated that the most significant difference in perfusion was observed in the anterior quadrangular lobule (H IV and V) on the left side (*p* < 0.0026). High differences in the median perfusion between the groups were also observed in a few other regions, with *p* < 0.05, but higher than premised *p* = 0.0026 (the Bonferroni correction was used in the statistical analysis). The assessment of the perfusion may be interpreted as a promising method of additional examination of atypical parkinsonisms with overlapping clinical manifestation, as in the case of PSP-P and MSA-P. The results obtained suggest that the interpretation of the differences in perfusion of the cerebellum should be made by evaluating the subregions of the cerebellum rather than the hemispheres. Further research is required.

## 1. Introduction

Multiple system atrophy (MSA) is an atypical parkinsonian syndrome neuropathologically defined as synucleinopathy [1,2] Clinically, it is characterized by parkinsonian symptoms accompanied by the parallel occurrence of cerebellar and autonomic dysfunction. Its diagnostic criteria have been revised recently [3]. Though the criteria evolve, there remains a lack of tools enabling a definite diagnosis of the disease. The clinical manifestation of MSA-Parkinsonian type (MSA-P) overlaps with other entities, especially Progressive Supranuclear Palsy-Parkinsonism Predominant (PSP-P) [4]. Both diseases may be associated with less pronounced cognitive deterioration and parkinsonian syndrome with a moderate response to levodopa treatment.

Progressive supranuclear palsy (PSP) is a tauopathic atypical parkinsonian syndrome with multiple phenotypes [5]. Among the phenotypes, the second most common—PSP-Parkinsonism Predominant—is one of the most difficult to examine due to its non-specific manifestation, often misdiagnosed as PD or MSA-P, especially in the early stages of the disease [6,7]. Another common feature is possible dysautonomia, the relatively slow progression or presence of speech and swallowing impairment [3,5]. Rapid eye movement (REM) sleep behavior disorder (RBD), traditionally associated with synucleinopathy, was recently described as less specific as it is present in up to 50–75% of PSP patients [8,9]. The clinical manifestation during the first years is rather inconclusive in terms of a differential diagnosis, as none of the early symptoms can be considered to be disease-specific. Moreover, the pathomechanisms of PSP and MSA are not fully understood. The contemporary literature lacks pathways explaining the course of the pathomechanisms leading to the different phenotypes of PSP. In PSP, among the hypotheses regarding its pathomechanism, astroglial and oligodendroglial tau accumulation and microglial activation [10] should be mentioned. In MSA the mechanism is associated with various possible mechanisms such as the increased protein intracellular accumulation of alfa synuclein, a consequence of the enhanced transcription and translation of the *SNCA gene* or neuroinflammatory activation, which has not been fully verified [11]. Most of the pathomechanisms of PSP and MSA are considered to be hypotheses rather than definite conclusions.

As ultimate confirmation requires post-mortem examination, a clinical diagnosis can only be probable at best and is often preceded by multiple additional tests among which neuroimaging, though being a secondary method in the criteria of diagnosis, seems to be one that is relatively easily accessible [5,12]. Magnetic resonance imaging is performed universally, but it is not the only possible method nor does it supply enough data. Currently, several papers describing the promising effects of imaging with the use of specific radiotracers dedicated to detect tau deposits, e.g., 18F-flortaucipir (AV-1451) or second-generation radiotracers such as ^18^F-PI2620, have been published [13,14,15]. However, this method is a high-cost procedure with low accessibility and off-binding artefacts [14,16]. Various studies have evaluated the significance of ^18^F-FDG PET, a positron emission tomography (PET) radiotracer, in the assessment of atypical parkinsonisms. MSA is associated with hypometabolism in the cerebellum, pons and putamen, whereas for PSP this occurs in the medial frontal cortices, prefrontal cortices, caudate nucleus, thalamus and midbrain [17,18,19]. In the context of PSP, the abnormalities of metabolism within the cingulate gyrus, the rostral cerebellum, the oculomotor vermis and the inferior parietal are associated with oculomotor dysfunction [20]. No studies were performed in the differential diagnosis of PSP-P and MSA-P using metabolic assessment in PET. The correlation of certain abnormalities in metabolic PET with oculomotor dysfunction, less commonly observed in PSP-P than in PSP-RS, shows the necessity for extending evaluation by indicating PSP subtypes. The application of perfusion single photon emission computed tomography (SPECT) was previously described in the context of the differential diagnosis of MSA-P and PSP-P [21,22], but in that analysis the cerebellum was not divided into smaller regions of interest (ROI). The presented analysis of perfusion SPECT in PSP-P and MSA-P in the cerebellum may be interpreted as an initial point for further discussion concerning whether the differentiation of MSA-P and PSP-P can be associated with a specific metabolic or perfusion profile in the cerebellum.

## 2. Method

Thirty-one right-handed patients aged from 50 to 79 participated in this study: 20 with MSA-P (14 females, 6 males) and 11 with PSP-P (7 females, 4 males). All the diagnoses were made in accordance with the diagnostic criteria current at the time of recruitment during hospitalization in the Department of Neurology, Medical University of Warsaw by neurologists experienced in the field of movement disorders [5,12]. After the publication of a revised version of the diagnostic criteria for MSA, the final study population was updated. The disease duration varied from 4–6 years. The severity of clinical symptoms and levodopa responsiveness was evaluated with the use of UPDRS-III. Scales dedicated to such specific entities as UMSARS or the PSP Rating Scale were not used in this study as the authors intended to use a test useable in both examined diseases—PSP-P and MSA-P. Regarding dopaminergic treatment, the patients received levodopa and amantadine. The levodopa doses varied from 0 to 1200 mg. Levodopa responsiveness assessed after 250 mg of levodopa intake was lower than 30% for every patient. The clinical characteristics of the analyzed population, including the L-dopa equivalent daily doses (LEDDs), are presented in Table 1. Every patient participating in the study underwent a brain and MRI scan (3T) up to 12 months prior to inclusion in the study. The final study population consists of patients who did not have any significant atrophy within the cerebellum (verified using volumetric analysis and compared with reference values from the literature).

Due to the different ages of disease onset in MSA and PSP, the groups could not be age-matched. The patients were recruited from January 2017 to December 2019. Each patient signed a written consent form before participation in the study. The exclusion criteria included coexistent cerebrovascular damage of the central nervous system (CNS), neoplasm and history of focal brain injury. The term “cerebrovascular damage” was interpreted as a previous history of stroke/TIA or the presence of a vascular lesion >1 mm detected in 3 T MRI.

### 2.1. SPECT

The SPECT examination was performed in the Department of Nuclear Medicine, Mazovian Brodno Hospital. The cerebral blood flow was assessed with the use of technetium-99m hexamethylpropyleneamine oxime ([99mTc]Tc-HMPAO). A dose of 740 mBq of [99mTc]Tc-HMPAO was administered to the patients in a quiet, dimly lit room. The acquisition was performed in a supine position with a SPECT/CT scan (Symbia T6, Siemens) on a dual-head gamma camera with a low-energy high-resolution parallel-hole collimator. A step-and-shoot acquisition mode was utilized. Sequences of 128 frames on a 128 × 128 matrix were used (64 projections per head, 30 s per projection). The photopeak was set at 140 keV with a 10% window each side. Iterative reconstruction (eight iterations, eight subsets, 7 mm Gauss filter), scatter correction and CT attenuation correction were performed. The post-processing examination was carried out using Scenium software (Siemens Medical Solutions USA, Inc., Malvern, PA, USA). The SPECT ROIs were preplanned in Scenium software (an integral part of the Siemens workstation) based on T1-weighted MRI images of a standard brain dataset.

The analysis and definition of subregions, including the subregions of the cerebellum, was based on the program offered by Siemens Healthineers (SCENIUM, Syngovia). The Scenium display and analysis software has been developed to aid the clinician in the assessment and quantification of pathologies taken from PET and SPECT scans. The software is deployed via medical imaging workplaces and is organized as a series of workflows which are specific to use with particular drug and disease combinations. The software aids in the assessment of human brain scans enabling automated analysis through the quantification of mean pixel values located within the standard regions of interest. It facilitates comparison with existing databases of healthy subjects and the reference parameters derived from these databases, which are derived from SPECT studies, e.g., the calculation of uptake ratios between regions of interest and subtraction between two functional scans. Cerebellar subregion segmentation was adopted from this generally accepted program. This program allows the 99mTc-HMPAO- SPECT/CT examination obtained from an individual patient to be superimposed on the database. The data-base contains the defined brain regions and sub-regions and the corresponding radiopharmaceutical accumulation values.

The databases were constructed from images for which Flash3D reconstruction and CT-based attenuation correction were performed and intensity normalization was based on the brainstem and the whole brain respectively. All of the databases are composed of HMPAO-SPECT scans of 20 asymptomatic control subjects with an age range of 64–86 years and a mixed population of males and females. HMPAO SPECT scans were performed on healthy volunteers as a part of a follow-up on asymptomatic controls in a dementia study. All the asymptomatic controls were given the same comprehensive examination, including medical and psychiatric examination, blood and CSF collection, ECG, CT, rCBF, orthostatic test and cognitive tests.

The regions of interest (ROI) used in Database Comparison are taken from Tzourio-Mazoyer et al. [23] and were defined on a high-resolution T1 MRI volume scan. It should be noted that the ROIs have some asymmetry. This is because they were defined on an actual subject [24].

In the edition of Database Comparison used, the statistics are computed and displayed on a voxel-by-voxel basis. The patient scan is first registered into the same space as the normal brain. For HMPAO-SPECT, an affine registration was calculated. The transformed scan was then smoothed using a 12 mm full-width half-maximum isotropic Gaussian kernel. Database Comparison then computes the number of standard deviations from the mean for each voxel, where the mean and standard deviation are taken from the corresponding voxel in the normal brain. According to the model, these statistics then follow a T-distribution.

The ROIs used in Database Comparison are predefined on the normal brain. By fusing the SPECT to the normal brain, Database Comparison can automatically position the ROIs on the patient brain. There are 134 separate ROIs that can be displayed and analyzed in Database Comparison. Database Comparison uses several fusion algorithms. A rigid fusion algorithm is used to fuse the SPECT scan to the supporting scan. An affine fusion algorithm is used to fuse the SPECT to the normal brain.

The study analysis is based on a comparison of the accumulation of the radiopharmaceutical in different regions of the brain. One of the adopted methods of analysis is to compare the accumulation of radiopharmaceuticals in individual ROIs in the subject with the database developed on the basis of studies in healthy people. The differences between the accumulation of the radiopharmaceutical in the selected ROI in the patient compared to the database are expressed in SD values. Statistically significant differences in the distribution of the radiopharmaceutical in the selected ROI are assumed if the accumulation exceeds the value of two standard deviations. This method was used in the SCENIUM program (Siemens). The analysis is based on the assessment of the value of the standard deviation. The total maximum and minimum counts were automatically measured in each ROI of the investigated brain SPECT and were compared using Scenium with measurements from the standard brain SPECT datasets. All the comparisons were automatically presented by Scenium as standard deviations. The values of standard deviations from the ROIs were evaluated in multiple locations in the brain by statistical analysis.

The shape and size of the SPECT examined brains were calibrated in accordance with the shape and size of the standard brains from the dataset. The pre-planned ROIs were then extrapolated to the SPECT images of the assessed brains. Finally, the total maximum and minimum counts were automatically evaluated in each ROI of the investigated brain SPECT and were differentiated using Scenium with measurements from the standard brain SPECT datasets. Indeed, using a smaller normalization region (e.g., cerebellum, brainstem) can improve the statistics in individual cases. However, there are several caveats that must be taken into account. In spite of a good registration of the patient’s large brain structures to the anatomical template, the small anatomical structures might not be in the correct position. This may lead to incorrect normalization ROI positioning and result in statistical errors. Therefore, particular attention was paid to the images obtained after normalization, verifying the results with the individual patient’s MRI image. In this study 19 different ROIs were analyzed (Table 2 and Table 3, Appendix A). The study did not include the assessment of any additional control group. The data were compared with a reference database comprising [99mTc]Tc-HMPAO brain scans of 20 healthy volunteers with an age range of 64–86 years (males and females). All the comparisons were automatically detected by Scenium as standard deviations. The values of the standard deviations from the ROIs were examined in multiple locations in the brain by statistical analysis. All the acquired data were evaluated by a specialist experienced in nuclear medicine.

### 2.2. Statistical Analysis

All analyses were performed using Statistica software (TIBCO Software Inc. (2017). Statistica (data analysis software system), version 13. http://statistica.io, version 13.1 Dell. Inc. Statsoft). The data distribution from the analyzed groups of patients (PSP-P and MSA-P patients) was assessed with the Shapiro–Wilk test. Due to non-normal distribution all the parameters are expressed as medians with a lower (Q1) and upper (Q3) quartile and their interquartile range (Q1–Q3). For the group comparison we used the Mann–Whitney U test. The significant results are presented as box plots. Due to multiple comparisons for a final decision in regard to statistical significance, a corrected *p*-value after Bonferroni correction was used. The calculated *p*-value of 0.0026 was considered significant.

## 3. Results

The detailed values of SPECT perfusion in regard to the analyzed parts of the cerebellum with given median values and lower and upper quartiles with their interquartile range (Q1–Q3) for the whole group, as well as for subgroups MSA-P and PSP-P patients, are presented in Table 2.

The largest difference between the median values of absolute SPECT perfusion between MSA-P and PSP-P patients was observed in the anterior quadrangular lobule (H IV and V) on the left side (MSA-P = −2.4 vs. PSP-P = 0.0). This difference was statistically significant (*p* = 0.0015, Table 3, Figure 1).

Also, a trend was detected between the median values in the analyzed subgroups (PSP-P and MSA-P patients) for the cerebellum and vermis (*p* = 0.0474), cerebellum 4 5 on the left and right (*p* = 0.0334 and *p* = 0.0285, respectively), and cerebellum 6 on the right side (*p* = 0.014), but higher than the corrected *p* = 0.0026, Table 3.

## 4. Discussion

To the best of our knowledge there have been no specific analyses of cerebellar abnormalities dedicated to the examination of the anterior quadrangular lobule in the context of the differential diagnosis between PSP-P and MSA-P prior to this study. Cerebellar atrophy and hypoperfusion are commonly interpreted as a feature of MSA [25]. Previous studies either did not provide an extended evaluation of the cerebellum or did not acknowledge PSP subtypes [21,25]. Cerebellar abnormalities are interpreted as one of the stages of the evolution of PSP. The atrophy within the cerebellum in PSP is usually observed after the initiation of morphological deterioration in the brainstem and subcortical nuclei [26]. The pathology of PSP within the cerebellum is associated with the dentate nuclei, but the majority of studies refer to PSP patients with neuropathological examination following PSP-RS [26]. In this context the sequence and correlation of the changes in PSP-P have been less explored, which may have resulted in independent pathophysiological sequences in the affected structures in the brain and cerebellum being proposed. This may be partly correlated with the fact that the perfusion of the cerebellum is a differentiating feature only in certain parts. Moreover, given that perfusion abnormalities are a stage preceding regional atrophic changes, the lack of significant changes in these regions of interest in the preliminary MRI is justified.

The interpretation of regions of hypoperfusion crops up in earlier studies regarding atrophies in neurodegenerative disorders. In a study evaluating atrophies in various entities, it was found that MSA should be associated with abnormalities in lobules I-IV [27]. Interestingly, the same regions were found to be deviated in PSP [27]. Neurodegenerative changes were also found to be disease-specific and corresponding in other diseases such as Alzheimer’s disease, amyotrophic lateral sclerosis and frontotemporal dementia [27]. The issue concerning the abnormalities of perfusion and metabolism in MSA generally concentrated on the global assessment of the cerebellum. In one work cerebellar hypoperfusion was associated primarily with MSA, whereas decreased perfusion within the thalamus was found in PSP [28]. The study did not provide a subdifferentiation of PSP phenotypes.

PSP-P is not commonly mentioned in the literature. The differentiation between MSA-P and PSP-P seems to be crucial in the context of their overlapping clinical manifestations. The neuroimaging parameters differentiating PSP-P and MSA-P were initially studied using MRI and SPECT. With regard to MRI, the Mesencephalon/Pons Ratio (M/P) and the Magnetic Resonance Parkinsonism Index (MRPI) were among the most relevant differentiating factors mentioned. Interestingly, the modification of MRPI—MRPI 2.0—which is feasible in the differentiation of PSP-P and PD, did not provide sufficient potential in the examination of PSP-P and MSA-P [25]. This may suggest that the evaluation of the ventricles, which are assessed in MRPI 2.0, may not be as beneficial as it initially appeared to be. Another paper indicates cerebellar hypoperfusion as a marker distinguishing MSA from PD and PSP-RS [24]. The lack of papers differentiating PSP-P and MSA-P is likely related to the fact that only the latest criteria for the diagnosis of PSP draw attention to a more precise examination of PSP-P. The fact that PSP-P is associated with up to 30% of PSP cases suggests the need for more detailed assessment [29]. In a paper evaluating iron accumulation in MRI in atypical parkinsonisms using R2* MRI it was found that significant differences in the red nucleus and the substantia nigra can be observed between PD, PSP, MSA and healthy controls, but not between PSP-RS and PSP-P. The authors did not perform an extended analysis of the differentiation of MSA and PSP-P [30].

Several papers describe the possible use of biochemical biomarkers in the differential diagnosis of parkinsonian syndromes. However, this approach lacks phenotype specificity, because the substances analyzed as potential marker candidates are usually connected with different pathology (e.g., alphasynucleinopathy vs. tauopathy) or, at the utmost, with specific disease, but without phenotype differentiation. In the case analyzed in the study, as PSP-P and MSA-P differ in the context of the underlying neuropathology, CSF or serum biomarkers could be also useful in differential diagnosis. The neurofilament light chain (NfL), phosphorylated neurofilament heavy chain (pNfH) and chitinase-3-like protein 1 (YLK-40) were described as statistically significant CSF differentiating biomarkers to discriminate tauopathies and synucleinopathies [31]. Low CSF coenzyme Q10 levels could be considered as a biomarker characteristic for MSA and could be useful in cases of PSP-MSA differentiation [32]. Currently, the development of serum-based tau or alpha-synuclein biomarkers is dissatisfactory, however high-throughput multiplex assays and panels are being developed and constantly evaluated in the context of possible clinical use [33]. Research concerning the biochemical markers of atypical parkinsonian syndromes is promising and its use could be beneficial, however it is currently limited by low accessibility in everyday clinical practice.

The main limitation of the study is the relatively small size of the population examined. This was due to the use of a one-center database and the rarity of the disease. The authors think that the analysis of a larger group could result in an increase of statistical significance of several regions which were identified as insignificant in the current manuscript due to the use of the Bonferroni correction. Another limitation is the lack of neuropathological confirmation of diagnosis which could not be performed as all participants remain alive. There is no additional control group as the software used in the study provided a comparison with a group of healthy volunteers aged matched with the examined group.

The assessment of perfusion may be interpreted as a promising method of additional examination of atypical parkinsonisms with overlapping clinical manifestation as in the case of PSP-P and MSA-P. The results obtained suggest that the interpretation of differences in perfusion of the cerebellum should be made by evaluating the subregions of the cerebellum rather than the hemispheres. Interestingly, the authors identified an asymmetrical significance among right-handed patients. The region differentiating PSP-P and MSA-P was found in the left cerebellar hemisphere. The results of this study are more of a preliminary exploration of this issue and further analyses based on larger groups of patients should be undertaken. More research in the field is required.

## Figures and Tables

**Figure 1 diagnostics-12-03022-f001:**
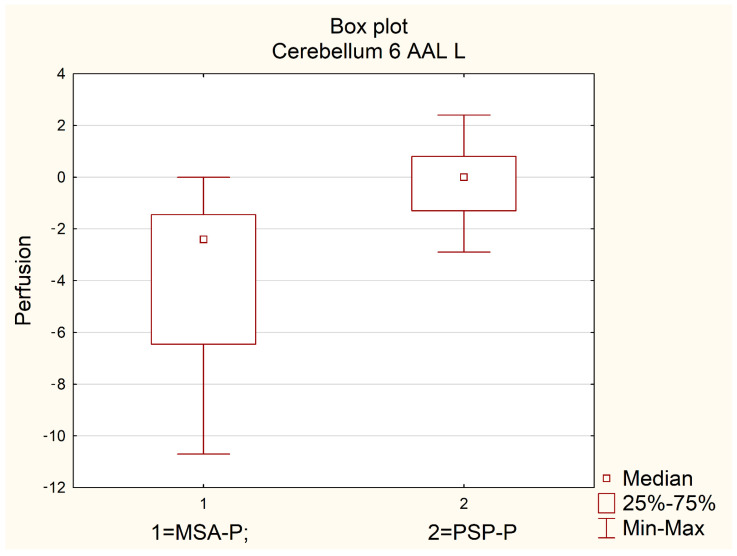
Comparison of perfusion in the left anterior quadrangular lobule (AAL L) in MSA-P and PSP-P.

**Table 1 diagnostics-12-03022-t001:** Clinical characteristics of the analyzed population.

	MSA-P	PSP-P
**age (mean)**	62.5 years	72 years
**disease duration**	4–6 years	4–6 years
**UPDRS part III ON state (mean)**	35 points	32 points
**UPDRS part III OFF state (mean)**	38 points	38.3 points
**Hoehn & Yahr (mean)**	2	2.5
**LEDD (mean)**	850 mg	1200 mg

**Table 2 diagnostics-12-03022-t002:** Descriptive statistics.

ROI—CEREBELLUM	Total (*n* = 31)				MSA-P (*n* = 20)				PSP-P (*n* = 11)			
	Median	Lower Quartile	Upper Quartile	Quartile Range	Median	Lower Quartile	Upper Quartile	Quartile Range	Median	Lower Quartile	Upper Quartile	Quartile Range
cerebellum + vermis SD	−1.60	−3.90	−0.10	3.80	−2.00	−4.00	−1.35	2.65	−0.50	−1.60	0.20	1.80
ant. quadrangular lobule (H IV-V) L SD	−1.20	−2.40	0.30	2.70	−1.30	−2.55	−0.05	2.50	−0.60	−1.60	0.50	2.10
ant. quadrangular lobule (H IV-V) R SD	−0.20	−1.70	0.50	2.20	−0.50	−2.00	0.70	2.70	0.00	−0.70	0.50	1.20
anterior quadrangular lobule 1 L SD	−1.60	−2.50	−0.10	2.40	−1.70	−2.75	−0.90	1.85	−1.30	−1.90	0.70	2.60
anterior quadrangular lobule 1 R SD	−0.90	−2.40	0.30	2.70	−1.30	−2.70	−0.35	2.35	0.20	−1.20	0.70	1.90
anterior quadrangular lobule 2 L SD	−2.10	−3.60	−0.50	3.10	−2.45	−4.35	−1.80	2.55	−1.00	−2.10	0.00	2.10
anterior quadrangular lobule 2 R SD	−1.50	−3.40	−0.60	2.80	−2.35	−3.95	−0.80	3.15	−0.70	−2.40	0.40	2.80
anterior quadrangular lobule (H IV and V) L SD	−1.90	−5.40	−0.10	5.30	−2.40	−6.45	−1.45	5.00	0.00	−1.30	0.80	2.10
anterior quadrangular lobule (H IV and V) R SD	−0.60	−3.30	0.60	3.90	−1.55	−3.95	0.35	4.30	0.50	−0.60	1.60	2.20
simple lobule (H VI-VII) L SD	−0.10	−1.00	1.20	2.20	−0.40	−1.55	0.70	2.25	0.10	−0.40	1.40	1.80
simple lobule (H VI-VII) R SD	−0.30	−1.10	1.00	2.10	−0.15	−0.85	1.35	2.20	−0.70	−1.30	0.20	1.50
cerebellar nuclei L SD	0.20	−0.70	1.40	2.10	−0.25	−1.20	0.65	1.85	1.50	−0.50	2.10	2.60
cerebellar nuclei R SD	0.50	−0.70	1.70	2.40	0.80	−0.80	1.45	2.25	0.30	−0.70	2.30	3.00
nodule (X) L SD	−2.90	−4.90	−1.20	3.70	−2.95	−5.00	−1.80	3.20	−2.80	−4.70	0.10	4.80
nodule (X) R SD	−2.90	−3.80	−0.70	3.10	−3.05	−3.70	−1.35	2.35	−1.00	−3.80	0.00	3.80
superior semiluminar lobule; first crus of antiform lobule (H VII a) L SD	−1.00	−1.70	0.20	1.90	−1.35	−1.85	0.20	2.05	−0.40	−1.50	0.20	1.70
superior semiluminar lobule; first crus of antiform lobule (H VII a) R SD	−0.30	−1.50	0.20	1.70	−0.20	−2.05	0.20	2.25	−0.50	−1.30	0.10	1.40
simple lobule (HVI) superior semiluminar lobule (first crus of antiform lobule (H VII) L SD	−0.90	−1.90	−0.10	1.80	−1.30	−2.10	0.10	2.20	−0.80	−1.80	−0.10	1.70
simple lobule (HVI) superior semiluminar lobule (first crus of antiform lobule (H VII) R SD	−0.70	−1.60	0.80	2.40	−0.30	−1.35	0.85	2.20	−1.10	−1.80	0.00	1.80

L—left; R—right; SD—standard deviation; ROI—region of interest.

**Table 3 diagnostics-12-03022-t003:** Comparison of MSA-P and PSP-P groups of patients in regard to cerebellum SPECT perfusion.

CEREBELLUM—ROI	P
cerebellum + vermis SD	0.0474
ant. quadrangular lobule (H IV-V) L SD	0.0863
ant. quadrangular lobule (H IV-V) R SD	0.3856
anterior quadrangular lobule 1 L SD	0.1423
anterior quadrangular lobule 1 R SD	0.0572
anterior quadrangular lobule 2 L SD	0.0334
anterior quadrangular lobule 2 R SD	0.0285
anterior quadrangular lobule (H IV and V) L SD	0.0015
anterior quadrangular lobule (H IV and V) R SD	0.014
simple lobule (H VI-VII) L SD	0.3626
simple lobule (H VI-VII) R SD	0.2309
cerebellar nuclei L SD	0.0601
cerebellar nuclei R SD	0.9341
nodule (X) L SD	0.342
nodule (X) R SD	0.1862
superior semiluminar lobule; first crus of antiform lobule (H VII a) L SD	0.397
superior semiluminar lobule; first crus of antiform lobule (H VII a) R SD	0.8685
simple lobule (HVI) superior semiluminar lobule (first crus of antiform lobule (H VII) L SD	0.6645
simple lobule (HVI) superior semiluminar lobule (first crus of antiform lobule (H VII) R SD	0.3112

## Data Availability

The data is available on request.

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
