# Peer review of "The Use of Cerebellar Hypoperfusion Assessment in the Differential Diagnosis of Multiple System Atrophy with Parkinsonism and Progressive Supranuclear Palsy-Parkinsonism Predominant"

_diagnostics, 2022, doi:10.3390/diagnostics12123022_

Round 1

Reviewer 1 Report

The manuscript “The use of cerebellar hypoperfusion assessment in the differential diagnosis of MSA-P and PSP-P” can be an interesting to understand the interpretation of differences in perfusion of the cerebellum. The manuscript has clear objectives and limitations too. Author have included 31 patients in total, which is not bad size for rare diseases. While outcome significance is marginally significant in the left cerebellar hemisphere. Which does not provide strong differentiating evidence for PSP-P and MSA-P. Authors are suggested to include the age matched controls in their study.

Author Response

Dear Reviewer 1, 

We are grateful for your valuable comments and suggestions. We believe that the current version of the manuscript significantly improved will meet the criteria for publication in Diagnostics. Please find below our responses to your comments. 

  1. While outcome significance is marginally significant in the left cerebellar hemisphere. Which does not provide strong differentiating evidence for PSP-P and MSA-P.

Due to the limitations of the study as the number of examined patients which is affected by the rarity of the diseases and the limited specificity of the assessment of perfusion SPECT, Authors used the Bonferroni correction to counteract multiple comparisons problem. The calculated p-value of 0.0026 was considered significant. The condition was met in one of the regions - anterior quadrangular lobule (H IV and V) L.

  1. Authors are suggested to include the age matched controls in their study.

The study is based on the analysis of the results compared with healthy controls, which is obtained using Siemens Scenium Software. In the methodological section authors indicated that:

All of the databases are composed of HMPAO-SPECT scans of 20 asymptomatic control subjects with an age range of 64-86 years, mixed population of male and female. HMPAO SPECT scans were performed on healthy volunteers as a part of a follow-up on asymptomatic controls in a dementia study. All asymptomatic controls were given the same comprehensive examination, including medical and psychiatric examination, blood and CSF collection, ECG, CT, rCBF, orthostatic test and cognitive tests.

Moreover the results of MSA-P and PSP-P patients, as mentioned in Table 2 are based on the mean standard deviations, which are a consequence of comparisons with healthy volunteers from the database. Due to the fact that the software already provides a comparison with healthy volunteers, Authors did not receive an approval for an analysis of an additional control group. The groups could not be fully age matched as MSA-P is a disease, which appear earlier than PSP-P.

Best regards
Natalia Madetko-Alster

Reviewer 2 Report

This is a SPECT study that uses cerebellar perfusion to distinguish ΜSA-P from PSP-P. The methodology is sound and the statistical analysis is adequate. The authors conclude that regional cerebellar perfusion measures may help in the differential diagnosis.This is an interesting study that tries to adress a clinical problem faced in everyday clinical practise. This study can be used as a reference point for future large scale studies.

I have some minor comments that i think will help improve the manuscript

1. The sentence in lines 39-40 is repeated a few lines later.

2. line 104 , how the lack of atrophy was assessed? volumetric analysis or radiologist opinion ?

3. Figure 1 presentation should be improved.

4. The authors could describe other potential biomarkers ( serum , CSF) that could help differentiate these conditions, besides imaging and describe their feasibility in clinical practise.

5. Disease onset was 4-6 years in both groups. Are there any inidications that these findings could be observed during the onset or perhaps preclinicaly in these conditions?

Author Response

Dear Reviewer 2, 

We are grateful for your valuable comments and suggestions. We believe that the current version of the manuscript significantly improved will meet the criteria for publication in Diagnostics. Please find below our responses to your comments. 

  1. The sentence in lines 39-40 is repeated a few lines later.

The change was implemented in the manuscript. The repetition was eliminated.

  1. line 104 , how the lack of atrophy was assessed? volumetric analysis or radiologist opinion

The atrophy was verified using volumetric analysis and compared with reference values from the literature. Due to the fact that the possible atrophic changes were verified in the context of their presence or absence

(the patients with present atrophic changes were not included in the study), authors did not perform an extended analysis of volumetric analysis of the structures. Authors fully agree that further analyses of the volumetric features would be interesting in further studies.

  1. Figure 1 presentation should be improved.

Figure 1 was improved.

  1. The authors could describe other potential biomarkers ( serum , CSF) that could help differentiate these conditions, besides imaging and describe their feasibility in clinical practise.

To the best of our knowledge there are no studies dedicated to the comparisons of serum and CSF parameters in the comparison of MSA-P and PSP-P. The published research concerning the differential diagnosis of PSP-P and MSA-P evaluated using PubMed database concentrates on imaging parameters as MRI and SPECT (verified by using phrases “PSP-P MSA-P”). Authors added information concerning several studies using biomarker using differentiation of PSP (without indicating subtypes) and MSA, however it is not verified whether the results of the studies would be confirmed in studies where PSP group would be bounded to PSP-P. A paragraph: “Several papers describe possible use of biochemical biomarkers in differential diagnosis of parkinsonian syndromes. However, this approach lacks phenotype specificity, as substances analyzed as potential marker candidates as usually connected with different pathology (e.g. alphasynucleinopathy vs tauopathy) or, at the utmost, with specific disease, but without phenotype differentiation. In case analyzed in the study, as PSP-P and MSA-P differ in the context of underlying neuropathology, CSF or serum biomarkers could be also useful in differential diagnosis. Neurofilament light chain (NfL), phosphorylated neurofilament heavy chain (pNfH), and chitinase-3-like protein 1 (YLK-40) were described as statistically significant CSF differentiating biomarkers to discriminate tauopathies and synucleinopathies [1]. Low CSF coenzyme Q10 levels could be considered as a biomarker characteristic for MSA and could be useful in case of PSP-MSA differentiation [2]. Currently, the development of serum based tau or alpha-synuclein biomarkers is dissatisfactory, however high-throughput multiplex assays and panels are being developed and constantly evaluated in the context of possible clinical use [3]. Research concerning biochemical markers of atypical parkinsonian syndromes is promising and its use could be beneficial, however it is currently limited by low accessibility in everyday clinical practice” was added to the discussion.

  1. Disease onset was 4-6 years in both groups. Are there any inidications that these findings could be observed during the onset or perhaps preclinicaly in these conditions?

The observations concerning the possible feasibility of SPECT in the differential diagnosis of PSP-P and MSA-P in their preclinical conditions would be interesting, however due to the clinical overlaps of PSP-P and MSA-P with PD in their preclinical and early disease stage, the diagnosis of PSP-P and MSA-P is affected by delays. While PSP-RS, due to its pronounced deficits often present in early stages as postural or oculomotor dysfunction, the manifestation of PSP-P is commonly misdiagnosed due to its resemblance with PD – PIGD type. Though we agree that the suggested observations concerning detailed SPECT analysis of the cerebellum in PSP-P and MSA-P in preclinical phase would be striking, obtaining such a study with a reliable number of examined patients would be difficult.

Round 2

Reviewer 1 Report

Thank you for the clarification.